# Antimicrobial Peptides towards Clinical Application—A Long History to Be Concluded

**DOI:** 10.3390/ijms25094870

**Published:** 2024-04-29

**Authors:** Laura Cresti, Giovanni Cappello, Alessandro Pini

**Affiliations:** 1Medical Biotechnology Department, University of Siena, Via A Moro 2, 53100 Siena, Italy; giovanni.cappello2@unisi.it (G.C.); pinia@unisi.it (A.P.); 2SetLance srl, Via Fiorentina 1, 53100 Siena, Italy; 3Laboratory of Clinical Pathology, Santa Maria alle Scotte University Hospital, 53100 Siena, Italy

**Keywords:** antimicrobial peptides, antimicrobial resistance, clinical development, preclinical development, chemical modifications, toxicity

## Abstract

Antimicrobial peptides (AMPs) are molecules with an amphipathic structure that enables them to interact with bacterial membranes. This interaction can lead to membrane crossing and disruption with pore formation, culminating in cell death. They are produced naturally in various organisms, including humans, animals, plants and microorganisms. In higher animals, they are part of the innate immune system, where they counteract infection by bacteria, fungi, viruses and parasites. AMPs can also be designed de novo by bioinformatic approaches or selected from combinatorial libraries, and then produced by chemical or recombinant procedures. Since their discovery, AMPs have aroused interest as potential antibiotics, although few have reached the market due to stability limits or toxicity. Here, we describe the development phase and a number of clinical trials of antimicrobial peptides. We also provide an update on AMPs in the pharmaceutical industry and an overall view of their therapeutic market. Modifications to peptide structures to improve stability in vivo and bioavailability are also described.

## 1. Introduction

Fleming’s discovery of penicillin marked the beginning of a new era in medicine, leading to the development of various classes of antibiotics. Antibiotics have revolutionized medicine by significantly reducing the mortality rates associated with bacterial infections [1]. They are a mainstay of modern healthcare, used to treat various bacterial illnesses, from minor infections to life-threatening diseases [2]. However, the overuse and misuse of antibiotics have led to the emergence of antibiotic resistance, one of the most pressing global health threats we face today [3]. Antimicrobial resistance (AMR) occurs when bacteria adapt and evolve, becoming resistant to the effects of antibiotics.

In February 2017, to focus and guide research and development related to new antibiotics, the World Health Organization (WHO) published its list of pathogens for which new antimicrobial development is urgently needed [4]. Within this broad list, 12 families of bacteria are highlighted as “antibiotic-resistant priority pathogens” are reported, including *Acinetobacter baumannii*, *Pseudomonas aeruginosa* and various *Enterobacteriaceae* They can cause severe and often deadly infections such as bloodstream infections and pneumonia [5].

The implications of AMR are profound. It reduces the effectiveness of antibiotics, leading to prolonged illnesses, increased mortality, and higher healthcare costs. Common infections that were once easily treated can become untreatable, posing a significant risk, especially to vulnerable populations, such as the elderly, young children, and those with weakened immune systems [6]. In addition, the economic cost of resistance to antibiotics is enormous. Antibiotic resistance is single-handedly killing more people than cancer and road accidents combined: 700,000 persons per year, and another 10 million are anticipated by the year 2050. The cost to the global economy is estimated at USD 100 trillion [7,8].

Since the turn of the 1990s, the development and commercialization of new antibiotics has slowed. There are few new antibiotics, and even fewer with new active ingredients [9]. Among new antimicrobial drugs, many are a combination of existing antibiotics or known antibiotics and other molecules (such as enzyme inhibitors). Recarbrio™ is a classic example, being a combination of carbapenem imipenem, the renal dehydropeptidase-I inhibitor cilastatin, and the novel β-lactamase inhibitor relebactam. Relebactam is a potent inhibitor of class A and class C β-lactamases, conferring imipenem activity against many imipenem-nonsusceptible strains. Recarbrio™ is approved in the USA and EU for the treatment of hospital-acquired bacterial pneumonia, ventilator-associated bacterial pneumonia in adults and other Gram-negative bacterial infections [10]. A combination of meropenem with vaborbactam (MER-VAB) has recently been reported by Duda-Madej et al. MER-VAB is a β-lactam/β-lactamase inhibitor combination approved by the FDA in 2017 for the treatment of urinary tract infections caused by MDR bacteria [11].

Global initiatives to deliver new antibacterial therapies or to complement alternative therapies are urgently needed. Research and innovation play a vital role in the fight against antibiotic resistance. Scientists are exploring alternative therapies, such as phage therapy, which uses viruses to target and destroy bacteria [12], new types of molecules that could have antimicrobial activity and new vaccines against superbugs [13]. There is also increasing emphasis on developing rapid diagnostic tools to enable healthcare providers to prescribe the most effective antibiotics promptly, reducing unnecessary or ineffective treatments [14].

A globally integrated strategy that includes different approaches (antibiotics, vaccines, diagnostics, antibodies, and new tools targeting the host, the microbiome, or delivered by phages) seems necessary to fight AMR effectively [15].

## 2. Antimicrobial Peptides (AMPs): An Overview

The widespread growth of resistance to traditional antibiotics worldwide has prompted a significant surge in research directed at introducing new and unconventional anti-infective medications to the market. Antimicrobial peptides (AMPs) have captured the attention of the scientific community, as demonstrated by a rapid increase in the number of articles published (Figure 1) [16].

AMPs are small bioactive proteins generally composed of 10–50 amino acids with a molecular weight of less than 10 KDa. Most AMPs are positively charged (2–13 net positive charges), derived primarily from lysine and arginine (a few may be histidine) in the sequence forming a specific cationic domain [17,18]. A few AMPs are negatively charged: examples are daptomycin and an antimicrobial neuroendocrine peptide called chrombacin, the latter bearing 12 net negative charges [19].

To emphasize the multifaceted nature of these molecules, the term “Host Defence Peptide” (HDP) [20,21] is now more commonly used to reflect the breadth of biological processes they influence, although the term AMP is still accurate in the case of activity against bacteria [22].

Gramicidin, the first AMP, was isolated in 1939 from a *Brevibacillus* soil bacterium and showed antibacterial activity against various Gram-positive bacteria in vitro and in vivo [23]. Others were subsequently isolated from bacteria, fungi and animals. Microorganisms produce them to kill other bacteria that compete for the same ecological niche. In plants and insects that do not have an immune system, they represent the primary defence against pathogens. Finally, in higher organisms like mammals, AMPs play roles as effectors in innate immunity, being primarily responsible for the direct inhibition of pathogens, while also modulating innate and adaptive immune responses [24,25,26].

Natural AMPs can be classified as bacteriophage/viral AMPs, bacterial AMPs, fungal AMPs, plant-derived AMPs and animal-derived AMPs on the basis of their origin [27]. Among them, insect-derived antimicrobial peptides have been widely studied. Some examples of insect AMPs are defensins, cecropins, proline-rich peptides and attacins. They have been identified in insect orders such as the Diptera, Hymenoptera, Hemiptera, Coleoptera and Lepidoptera [28]. Focusing on mammalian-derived AMPs, the two major families are human cathelicidins and defensins, mainly produced by epithelial cells and neutrophils [29,30].

The large Antimicrobial Peptide Database (APD3; available at https://aps.unmc.edu/, accessed on 1 March 2024) [31] contains 3940 antimicrobial peptides including 3146 natural antimicrobial peptides (AMPs) from the six kingdoms (383 bacteriocins/peptide antibiotics from bacteria, 5 from archaea, 8 from protists, 29 from fungi, 250 from plants and 2463 from animals), 190 predicted and 314 synthetic AMPs (last updated: January 2024|Copyright 2003–2024 Dept of Pathology & Microbiology, UNMC). They have a variety of biological activities, such as antibacterial, antiviral, anti-cancer, immune regulation, wound healing and antioxidant properties, and can prevent and/or eradicate biofilms [32].

So far, we have considered natural AMPs; synthetic AMPs are also possible and can be obtained by various methods. One strategy used to develop new synthetic AMPs is the de novo design of new sequences aided by specific software [33]. The rational or computational design of new peptides has been used for several years, but Artificial Intelligence and Machine Learning applications (AI/ML) are fundamentally revolutionising the drug development process, including research in the field of peptides [34]. Lin et al. describe a new de novo-designed peptide selected by an Artificial Intelligence (AI) AMP classifier with strong antibacterial activity against antibiotic-resistant bacterial strains [33]. Various AI tools and platforms can be involved in different stages of this process. Szymczak et al. present an interesting and exhaustive analysis of the AI methods that could support AMP discovery and design, discussing different categories of AI methodologies and focusing on the recent achievements in AI-driven AMP discovery [35]. Since the de novo design of antimicrobial peptides is a complex task, a multidisciplinary approach involving expertise in biology, chemistry and bioinformatics is essential for success [36].

A completely different way to identify new sequences is the phage display technique. Phage display is a highly effective and robust technology used to identify ligands of biological targets, first reported by Smith in 1985 [37]. Phage display is a selection technique in which a library of peptide or protein variants is expressed on the outside of a phage virion (i.e., M13 phage), while the genetic material encoding the variants resides on the inside. This creates a physical link between variant protein sequences and the DNA encoding them, which allows rapid partitioning based on binding affinity to target molecules (antibodies, enzymes, cell-surface receptors, etc.) by an in vitro selection process called panning. Briefly, panning is carried out by incubating a library of phage-displayed peptides with the target, washing away the unbound phages, and eluting the specifically bound phages. The eluted phages are then amplified and taken through additional binding/amplification cycles to enrich the pool in favour of binding sequences. After 3–4 rounds, individual clones are characterised by DNA-sequencing and -binding assays. This technique has allowed the discovery of several new peptide candidates [38].

### Peptide Production

Extracting and purifying natural antimicrobial peptides (AMPs) from animals and plants poses challenges. The recent application of genetic engineering technology to the recombinant expression of AMPs has emerged as a significant development in their commercial production. AMPs are commonly expressed using prokaryotic or eukaryotic systems. Prokaryotic expression offers advantages such as short expression periods, high quantity and low cost, but challenges may arise during purification due to a lack of post-translational modification. On the other hand, eukaryotic expression offers advantages such as non-toxicity to host cells and simplified purification through extracellular expression, with drawbacks including higher expression costs and longer expression periods [39].

*E. coli* expression is a widely used prokaryotic expression system in the field of genetic engineering to industrialise antimicrobial peptide production [40]. Yeast expression systems are also possible such as with *Saccharomyces cerevisiae* and *Pichia pastoris* [41]. Both expression systems have negative and positive aspects at the same time, regarding expression efficiency, timing of procedures, expression costs and effects of non-biological activity. In recent years, researchers have used transgenic technology to introduce antimicrobial peptide (AMP) genes into plants like corn and soybean [42]. Plants, as advanced eukaryotes, offer a suitable environment for expressing AMPs. However, few studies have focused on extracting AMPs from plants used as expression hosts, so more work is needed to develop a fully functional expression system [43].

Another method for obtaining AMPs is chemical synthesis in the laboratory using an automatic synthesiser. The chemical synthesis of peptides is well developed, particularly solid-phase peptide synthesis (SPPS) described by Merrifield in 1963 [44]. SPPS technology has undergone significant enhancements and plays a pivotal role in contemporary peptide production. It is based on the coupling and deprotection of amino acids in a single reactor, leading to the development of automated peptide synthesisers. In comparison to recombinant technology, SPPS yields crude peptides that are more homogeneous, avoiding additional biological compounds. Impurities in the final SPPS product are easily identified, originating primarily from incomplete or side reactions during synthesis; this simplifies subsequent purification [45]. After synthesis, the purity of the peptide is assessed by techniques like mass spectrometry, nuclear magnetic resonance (NMR) and high-performance liquid chromatography (HPLC).

## 3. AMP-Based Drug Discovery

Drug development is a complex, lengthy and expensive process that begins with the design, synthesis and optimisation of a therapeutic compound [46]. In an initial phase of development, one or more antimicrobial compounds are identified by different techniques and may be of natural or synthetic origin. An important phase is the establishment and scale-up of the manufacturing process. In the preclinical phase, various in vitro tests are performed to study different aspects of the molecule, including its physiochemical properties, potency/cell activity (e.g., minimum inhibitory concentration and minimum bactericidal concentration), mechanism of action, efficacy and toxicity (especially in human cell lines), mechanisms of resistance, genotoxicity and interaction. A molecule can undergo an optimisation process aimed at improving aspects, such as affinity for the target, efficacy or toxicity. In some cases, artificial models can be of help for better characterisation before in vivo analyses or for minimising animal use. Rolhion et al. used an artificial gastrointestinal (GI) system to assess the effects of bacteriocin (Lmo2776) on the human gut microbiota composition in vitro [47]. Then, an in vivo model must be set up in different animal species in order to evaluate the safety and efficacy of potential drug candidates. This is accomplished using codified animal models and validated procedures. The ultimate goal is to translate the animal model responses into an understanding of the risk for human subjects. To this end, the toxicologist must be aware of the international guidelines for safety evaluation in humans. The typical toxicology profile consists of safety pharmacology, genetic toxicology, acute and subchronic toxicology, chronic toxicology, absorption, distribution, metabolism and excretion (ADME) studies, reproductive and developmental toxicology, and an evaluation of carcinogenic potential [48,49]. If the outcome of the preclinical phase is positive, the molecule officially enters clinical development. In clinical phase I, tolerability and potential dosing are assessed in a small number of healthy volunteers. In clinical phase II, a study with a small group of volunteer patients who have relevant infections is conducted to demonstrate the molecule’s positive impact. In clinical phase III, the medicine is tested in large, randomised, placebo-controlled trials with larger numbers of volunteer patients to confirm the efficacy and safety profile by generating statistically significant data. The clinical trials from all phases deliver the data required to prepare submissions for regulatory approval to agencies around the world. Teams of various disciplines (e.g., scientists, physicians, pharmacologists) need to work together until a new biopharmaceutical is finally developed and approved for use in humans by the regulatory agencies (e.g., EMA, European Medicines Agency; FDA, U.S. Food and Drug Administration) [50,51].

The duration of the process is influenced by various factors and may vary widely. The entire development cycle, from discovery to market release, typically takes 8 to 18 years, with an average duration of 13 to 14 years. The cost per molecule/candidate, measured in million euros (m€), does not include additional costs related to attrition (failed programs) and missed opportunities resulting from prolonged cycle times before advancing to the next developmental phase. Such extensions may increase the budget requirements for the initial stages by up to 50–100 m€ (Figure 2) [52].

## 4. Peptides in the Pharmaceutical Industry

Peptide drugs account for about 5–6% of the global pharmaceutical market, with impressive global sales reported in the recently updated “Global Peptide Therapeutics Market (by Type, Synthesis Technology, Manufacturing Type, Application & Region): Insights & Forecast with Potential Impact of COVID-19 (2022–2026)” [53]. The global peptide therapeutics market is expected to record a value of USD 44.43 billion in 2026, progressing at a compound annual growth rate (CAGR) of 6.9%, over the period 2022–2026. To further confirm these data, the “Peptide Therapeutics Market by Application, by Route of Administration, by Distribution Channel: Global Opportunity Analysis and Industry Forecast, 2021–2031” [53] reports that the peptide therapeutics market is estimated to reach USD 64.3 billion by 2031, growing at a CAGR of 6.8% from 2022 to 2031 (Figure 3A). The fastest-growing regional market is North America due to increasing R&D towards innovative peptide therapeutics and improvements in healthcare infrastructure, coupled with the growing prescription of peptide therapeutics on account of the high frequency of patients with chronic diseases such as diabetes and cancer. The global peptide therapeutics market is led by international companies, including Takeda Pharmaceutical (Tokyo, Japan), Pfizer (New York, NY, USA), Merck & Co. (Rahway, NJ, USA), Eli Lilly and Company (Indianapolis, IN, USA), Sanofi S.A. (Paris, France), AstraZeneca plc (Cambridge, UK) and GlaxoSmithKline (Brentford, UK). Over the years, the demand for peptide therapeutics has increased significantly: 114 peptides have already been approved by the regulatory authorities as therapeutic agents (Figure 3B) [54]. Their therapeutic indications include cancer, inflammatory, autoimmune and metabolic diseases and microbial infections. The latter may be promising new antibiotics or auxiliaries for traditional antibiotic therapy [55,56,57]. Another interesting and very promising aspect is their cell-penetrating properties. Cell-penetrating peptides are now under study as drug delivery tools for anti-cancer, antibacterial and antiviral therapies [58].

## 5. Therapeutic AMPs on the Market

Bacitracin is a group of cyclic polypeptides produced by organisms of the licheniformis group of *Bacillus subtilis*. It has antimicrobial activity against many Gram-positive bacteria, including *Staphylococci*, *Streptococci* and *Clostridia*, and was approved by the FDA in 1948 [59]. It is applied topically to treat local infections, mainly infections of the skin, ear and eye. 

Colistin (Polymyxin E) and Polymyxin B (PMB) belong to the Polymyxin class and are lipopeptide antibiotics with activity against many Gram-negative bacteria. The Polymyxins were approved for clinical use in the late 1950s but fell out of favour in the mid-1970s due to concerns about their potential nephrotoxicity and neurotoxicity [60]. They have since undergone many interesting modifications, not least their conjugation with already known antibiotics (e.g., a clinical trial of colistin–rifampicin administered i.v. [61]) with new approvals by the FDA (e.g., colistimethate sodium, a form of colistin, was approved in 1999).

Daptomycin is a lipopeptide isolated in the 1980s with impressive activity against Gram-positive, but not Gram-negative, bacteria [62]. It received approval by the US FDA in 2003 and it is currently widely used to treat *Staphylococcus* spp. and *Enterococcus* spp. infections [63].

Vancomycin is a tricyclic glycopeptide which acts against Gram-positive bacteria, including methicillin-resistant strains of *Staphylococcus aureus*, by inhibiting the synthesis of the peptidoglycan layer of the bacterial cell wall. It was first approved for use in the United States in 1958 and continues to be widely used, particularly with the recent rise in the incidence of serious MRSA infections [64].

Dalbavancin, Oritavancin and Telavancin are small lipoglycopeptides derived from vancomycin. They are more potent and bactericidal than their prototype and effective against vancomycin-resistant bacteria. They inhibit bacterial cell wall formation, and Telavancin and Oritavancin also disrupt bacterial cell membranes and affect membrane permeability. They were approved by the FDA between 2009 and 2014 [65].

Teicoplanin is a glycopeptide produced by *Actinoplanes teichomiceticus*, effective against Gram-positive bacteria resistant to β-lactam antibiotics. It has been used clinically for the treatment of methicillin-resistant *S. aureus* (MRSA) infections. Teicoplanin is not yet approved by the FDA for use in the USA but is widely used in Europe, Asia and South America. Some recent studies have documented its activity against SARS-CoV-2, reporting it as a possible drug of choice in the treatment of COVID-19 patients. Teicoplanin arrests the replication of the virus while preventing the development of Gram-positive bacterial co-infections [66].

Gramicidin, derived from the soil bacterium *Bacillus brevis*, is active against most Gram-positive and a few Gram-negative bacteria and fungi. It is often formulated with other active ingredients in topical creams, lotions and powders, topical and ophthalmic ointments, and ophthalmic and otic solutions. Gramicidin is unsuitable for systemic use due to its toxicity [67].

## 6. AMPs in Clinical Development

As in the case of many other classes of drugs, the number of AMPs entering clinical trials is significantly lower than the total number of compounds initially identified, and the percentage of those receiving marketing approval is even lower. The drug discovery process is conventionally represented as a funnel, rendering the idea of the falloff in the number of candidates phase by phase. Various AMPs are undergoing clinical trials for the prevention and treatment of different infections.

The landscape of drug development is dynamic, and the status of specific compounds is constantly evolving. For the latest information on AMPs in clinical phases, including their current development status, it is recommended to check the updated clinical trial databases. However, information on some peptides cannot be obtained if licences are transferred to other companies or preclinical or clinical trials are discontinued for unknown reasons.

A short description of some AMPs currently in clinical trials follows.

hLF1-11 is a short synthetic peptide derived from the N-terminal region of human lactoferrin, an antimicrobial protein found in human milk and other body fluids. Interestingly, hLF1-11 shows poor antimicrobial activity under physiological conditions in vitro, but has highly effective in vivo activity against bacteria (Gram positive and negative) and fungi, including infections caused by methicillin-resistant *S. aureus* (MRSA), *Klebsiella pneumoniae* and *Listeria monocytogenes*. hLF1-11 also shows immunomodulatory activity, stimulating monocyte differentiation and the release of pro-inflammatory cytokines [68].

EA-230 is a newly developed synthetic linear tetrapeptide derived from human chorionic gonadotropin. EA-230 exerted immunomodulatory and renoprotective effects in preclinical models [69] and its safety and efficacy have been demonstrated in phase I and II clinical trials [70].

The specifically targeted antimicrobial peptide (STAMP) C16G2 was developed to target mutants of the cariogenic oral pathogen, *Streptococcus* spp. [71]. *Streptococcus* spp. mutants are believed to be a critical factor in dental caries or tooth decay. C16G2 is being developed for the prevention of dental caries in adults, adolescents and children. C3 Jian, Inc., a biotechnology company focused on reengineering the human microbiome to deliver novel healthcare products, started a randomised, double-blind phase II clinical study in healthy adult subjects, with preliminary data released in 2015 [72]. In 2022, Namburu et al. reported that C16G2 was recognised by the FDA as an investigational drug for the prevention of dental caries, and it has efficiently concluded phase II clinical trials [73].

NP213 (Novexatin^®^) is a novel antifungal peptide specifically designed for the topical treatment of onychomycosis. NP213 was designed using Host Defence Peptides (HDPs), essential components of the innate immune response to infection, as a template. It was effective in two phase IIa human trials, confirming its promise as a peptide-based candidate for the topical treatment of fungal infections of the skin [74].

Dusquetide (SGX942) is a first-in-class Innate Defence Regulator (IDR) that modulates the innate immune response to PAMPs and DAMPs by binding to p62, a key adaptor protein that functions downstream of the key sensing receptors (e.g., Toll-like receptors) that trigger innate immune activation. There are no other candidates that target the p62 protein [75]. A phase III clinical trial, sponsored by Soligenix Inc. for the treatment of oral mucositis, a side effect of treatment of squamous cell carcinoma of the oral cavity, is ongoing [76].

Omiganan (CLS001) is an AMP analogous to indolicidin, a bovine member of the cathelicidin family. It showed antibacterial and antifungal activity in a range of preclinical and clinical studies, with a good safety profile. Although the phase IIIb trial for catheter-associated urinary tract infection caused by *S. aureus* failed [77], the phase III trial for the treatment of topical skin antisepsis and rosacea [78] and the phase II trial for vulvar intraepithelial neoplasia, acne, and atopic dermatitis are still ongoing [79].

Ramoplanin (NTI-851), sponsored by Nanotherapeutics, is a peptide produced by *Actinoplanes* spp. that exhibits bactericidal activity by blocking the cell wall peptidoglycan synthesis of Gram-positive bacteria. It is in phase III clinical trials for the oral treatment of vancomycin-resistant Enterococcus (VRE) infection and in phase II clinical trials for the treatment of *Clostridium difficile* [80].

Nisin is approved as a food preservative by regulators in over 80 countries, including the Food and Drug Administration (FDA) and the European Food Safety Authority (EFSA) [81]. Moreover, this AMP is under evaluation in some clinical trials for its application in the treatment of oral cavity squamous cell carcinoma [82].

## 7. Clinical Applications of AMPs—Challenge and Strategies

The development of AMPs for clinical application faces various challenges, including the high costs of development and production, reduced efficacy in clinically significant settings, and the unexpected emergence of bacterial resistance. AMPs act mainly on membranes but are not completely selective of microbial cells and may also be toxic to eukaryotic cells. Several AMPs cause haemolytic and/or cytotoxic effects at antimicrobial concentrations, limiting their wider utilisation [83]. Another drawback for clinical development is the lower antimicrobial activity in clinical environments [84]. AMPs may lose their bactericidal activity under physiological saline conditions due to a loss of electrostatic interactions with cell membranes. In the presence of serum, AMPs may bind to proteins such as albumin [85]. They may also be susceptible to proteolytic degradation [86]. Various strategies have been implemented to overcome some of these limitations, improve the performance of AMPs and render them suitable for clinical use.

### 7.1. Resistance to AMPs

Unlike traditional antibiotics, which work by rupturing the cell envelope or interrupting DNA replication or protein synthesis [87], some AMPs have multiple modes of action with non-specific targets and are less liable to develop resistance [88]. Some organisms produce several different AMPs, which may also reduce the likelihood of resistance development [89]. For these and other reasons, it was erroneously believed that resistance to AMPs was very unlikely to arise and therefore not a big concern [90]. However, as recent studies have shown, resistance to AMPs may not only be an intrinsic mechanism but may also be acquired or evolve at high rates (at least in vitro), generating mutants sometimes with high levels of resistance [91].

Intrinsic resistance to AMPs may be passive or inducible. Passive AMP resistance occurs in bacterial species such as *Proteus*, *Morganella*, *Providencia*, *Serratia* and *Burkholderia* as a result of an inherently more positively charged lipid A that reduces AMP interaction. The induction of AMP resistance in other bacteria is closely linked to environmental conditions and is a mechanism for bacterial survival in natural environments where they could be threatened by AMPs [92]. AMP resistance [87,88] may be caused by proteolytic degradation, a bacterial membrane-targeting impediment by secreted bacterial proteins, outer membrane vesicles (OMVs) or capsules; activation of the bacterial efflux pump; or decreased net anionic charge in the cell envelope [92].

As regards acquired resistance, Liu et al. reported a case of resistance to AMPs due to horizontal gene transfer in *E. coli* [93]. A plasmid containing the *mcr-1* gene was shown to mediate colistin resistance by encoding a phosphoethanolamine transferase that modifies lipid A, reducing its negative charge. The reduced affinity between colistin and LPS anchored to the bacterial outer membrane decreases the efficacy of colistin in clinical practice [94,95]. The *mcr-1*-containing plasmid was initially isolated in Chinese livestock, and after its initial characterisation, it was identified retroactively in 3 out of 1267 human faecal microbiome samples from China prior to 2011, indicating animal-to-human gene transfer [96].

So while current understanding suggests that most AMPs are generally less likely to develop drug resistance than traditional antibiotics, it is important to be aware of this possibility in clinical and environmental settings.

### 7.2. Chemical Modifications

Since structure and activity are related, the sequence, position and configuration of AMP amino acids play an important role in the biological activity of the peptide. Several chemical modifications may improve certain properties (i.e., antibacterial activity, permeability), making the AMP more stable and therefore more suitable for use in vivo. Some possible modifications are as follows.

#### 7.2.1. PEGylation

The term PEGylation describes the modification of biological molecules by covalent conjugation with polyethylene glycol (PEG) (Figure 4A), a non-toxic, non-immunogenic polymer. It is used as a strategy to overcome some disadvantages associated with molecules of interest [97]. The advantage of PEG residues is their very good solubility in aqueous and organic environments, great flexibility, high hydration that increases their hydrodynamic volume, and a range of molecular weights with low polydispersity. All these properties are acquired by compounds to which PEG is bound covalently. Proteins conjugated with PEG exhibit increased solubility and become resistant to antibodies, proteolytic enzymes and cells; because of their increased size, they are ultra-filtered more slowly by the kidneys [97]. So PEGylation of AMPs enhances their overall pharmacodynamic properties. Since the introduction of the first PEGylated protein, Adagen^®^, in 1990, an increasing number of pegylated products has appeared on the market [98].

#### 7.2.2. Lipidation

Another strategy to enhance the antimicrobial power of AMPs without modifying their properties is lipidation, namely the attachment of a portion of a fatty acid to N-terminal residues or lysine side chains (Figure 4B). The incorporation of lipid tails of different lengths enhances AMP hydrophobicity and improves membrane interaction, permeability and protection against proteolysis by enzymes [99]. The improved power is presumably correlated with the length of the acyl chain, which also influences AMP specificity and enhances interactions between the bacterial cell membrane and the fatty acid conjugated with the peptide [100]. However, as acyl chains grow in length, they increasingly tend to self-assemble in aqueous solutions, which causes a loss of peptide interaction with bacterial membranes. The length of the conjugated fatty acids may also increase hydrophobicity, enhancing selectivity for mammalian cells with consequent toxicity. So although an increase in the hydrophobicity of peptides can improve antimicrobial activity, it is crucial to preserve the right hydrophilicity/hydrophobicity balance to avoid an increase in toxicity. Clearly, a well-chosen chain length is key to determining the balance between improved antibacterial properties and selectivity [101,102].

#### 7.2.3. Cyclisation

Another covalent modification of the structure that could have an important effect on the function and consequently on the activity of AMPs is cyclisation (Figure 4C). Peptide cyclisation is a widely recognized strategy for enhancing serum stability. This is attributed to the increased volume resulting from cyclisation, which reduces the probability of protease contact through steric hindrance [19]. CP-11 is a well-known peptide analogue of indolicidin, and its cyclic form “cycloCP-11” has been reported [19,103]. Its antibacterial activity is similar to that of the linear form, but cyclisation greatly increased its stability to proteases. These findings suggest that cyclisation may be an important strategy in the rational design of antimicrobial peptides.

#### 7.2.4. D-amino Acids

Natural AMPs are composed of L-amino acids. Replacing L-amino acids with their D-stereoisomers results in peptides that are not recognised by naturally occurring proteases or immune system receptors due to their spatial configuration, and this is a strategy used to overcome the problem of biostability in vivo [104]. Natural amino acids in AMPs are easily recognised by the host proteolytic enzyme, leading to proteolysis of the peptide. Introducing D-amino acids at the site of the proteolytic cleavage interferes with this recognition and avoids peptide degradation [105]. The D-amino acid substitution strategy is also used to enhance peptide stability. The isomerisation of AMP amino acids also broadens their antimicrobial spectrum to Gram-positive bacteria. SET-M33 antimicrobial peptides have proven to be very active against Gram-negative bacteria. The isomeric version synthesised with D-amino acids showed 4- to 16-fold higher activity against Gram-positive pathogens, including *S. aureus* and *S. epidermidis*, than the peptide with L-amino acids. The antimicrobial activity of both peptide isoforms is influenced by their differential sensitivity to bacterial proteases [106,107]. Falciani et al. [106] and Brunetti et al. [107] reported that the antimicrobial peptide SET-M33 synthesised with D- instead of L-amino acids killed Gram-positive bacteria in vitro and in vivo because of its greater resistance to bacterial proteases.

#### 7.2.5. Branched AMPs

As already mentioned, the use of peptides as therapeutic agents has been limited by their short half-life in vivo. Since peptides are mainly broken down by proteases and peptidases, peptide stability is a bottleneck in the development of new peptide-based drugs. Multiple-Antigen Peptides (MAPs) are bioactive peptides synthesised in a branched form with a peptidyl core of radially branched lysine residues that covalently bind more copies of the linear sequence of the same peptide. In this way, one lysine can allow the synthesis of a two-branched peptide, three lysines, a tetra-branched peptide, and so on (Figure 4D).

The solid-phase synthesis of branched peptides was first described by Tam in the 1980s [108] and is based on the concept of using trifunctional amino acids to construct branched peptide-based molecules that he named MAPs. The first idea was to use these molecules to obtain synthetic vaccines, though different applications followed this first idea. It is now known that bioactive peptides become more resistant to the proteolytic activity of plasma and serum enzymes when synthesised in MAP form. Peptide biological activity can even be enhanced by multimeric binding. Monomeric peptides are cleaved very rapidly by peptidases into inactive peptides. Peptidases acting on small peptides are mainly Zn metallopeptidases and their catalytic site is located in a deep channel accessed only by small peptides. The steric hindrance of branched peptides may limit their access to the cleavage site, prolonging peptide half-life and improving their fitness for use as drugs. In addition, the enhanced efficacy of branched peptides in vitro and in vivo is generally ascribed to their multimeric nature, which promises a greater number of interactions than the corresponding monomeric form [106,109,110].

An interesting case of a peptide synthesised in MAP form is SET-M33. This synthetic antimicrobial peptide has been patented by SetLance srl, a biopharmaceutical company based in Siena, and is synthesised in the laboratories of the University of Siena. SET-M33 is a non-natural peptide synthesised in a tetra-branched form that makes it more stable in biological fluids. SET-M33 has shown high antimicrobial activity in vitro and in vivo, anti-inflammatory activity, a lack of immunogenicity and an ability to eradicate biofilms [111,112,113].

### 7.3. Toxicity of AMPs

Toxicity, together with instability and short half-life in vivo, is a major challenge for the clinical application of AMPs. In vitro and in vivo toxicity evaluation is essential in the development of new drugs [114]. Different in vitro and in vivo tests are required, including the EC_50_, which is the peptide concentration required to kill half the cells in vitro, and LD_50_, which is the dose required to kill half the animals in an in vivo experiment [115]. 

A major effect of AMPs is haemolysis. The main theory used to explain the mechanism of haemolysis is based on the cationic and amphiphilic structures of AMPs, together with peptide length, special amino acids and peptide chain helicity. Like membrane-breaking by pathogens, natural AMPs with cationic charges can interact with negative ions on the erythrocyte surface, forming oligomers that destroy the cells [116,117].

A well-known adverse effect of AMPs is their renal toxicity in vivo when administered parenterally. The nephrotoxicity of peptides such as vancomycin [118] and Polymyxin [119], already in clinical use, is well known in the literature. Acute kidney injury (AKI) occurs in up to 50%–60% of patients receiving Polymyxin and this aspect is the major dose-limiting adverse effect of Polymyxins [120]. The plasma concentrations of peptides associated with the increased risk of AKI overlap those required for antibacterial effect, making the therapeutic window narrow [121]. Studies conducted in cell lines and preclinical models in vivo show that AMPs may generally be toxic to renal tubule cells. The cell mechanisms involved include oxidative stress, apoptosis (via mitochondrial, death receptor, and endoplasmic reticulum pathways), cell cycle arrest, and autophagy [122,123,124,125]. Many AMPs in clinical and preclinical development have safety pharmacology data that demonstrate a certain level of nephrotoxicity in different animal models. Cresti et al. [125] reported that clinical laboratory investigations in dogs and rats showed a dose-related increase in creatinine and urea levels following intravenous administration of the SET-M33 peptide. These values, together with necropsy studies of animal tissues, indicate that kidneys are the target organ [125].

Dose-dependent nephrotoxicity occurring after intravenous administration is the major limiting factor for dose escalation. On the other hand, inhaled AMPs can achieve higher drug exposure directly at the site of infection, for example, in the lungs, while minimising systemic drug exposure. However, adverse effects on the lung caused by inhaled therapy have also often been reported [126,127,128].

### 7.4. Nanocarriers: A Strategy to Overcome AMP Toxicity

Since the antimicrobial mechanism also determines a certain level of toxicity, the therapeutic window of many AMPs is particularly narrow. In order to overcome this issue, many strategies have been tried. One is based on the encapsulation of AMPs in biocompatible nanocarriers, as widely reported in the literature, in order to reduce local toxicity while maintaining efficacy. Nanoparticles (NPs) may be classified as inorganic or organic. Inorganic NPs are generally composed of metals, such as iron (Fe), aluminium (Al), silver (Ag) or gold (Au). Gold NPs (AuNPs) are considered among the most biocompatible inorganic nanosystems and suitable for coupling active molecules, such as antimicrobial peptides [129]. Casciaro et al. designed a system named AuNPs@Esc(1–21) where gold-nanoparticles are coated with Esculentin-1a(1–21), an antimicrobial peptide derived from frog skin, for use as a drug against infections caused by *P. aeruginosa* [130]. Subaer et al. reported the powerful antimicrobial potential of the LL-37 peptide conjugated with AuNPs (LL-37@AuNPs) [131]. Regarding organic NPs, various systems, like liposomes, lipid-based nanoparticles, polymeric micelles and polymeric nanoparticles, have been reported in the literature [132]. Today, poly(lactic-co-glycolic acid) (PLGA), an aliphatic polymer with a polyester structure that is formed by copolymerisation of poly-lactic acid (PLA) and poly-glycolic acid (PGA), is the most widely used polymer. PLGA has interesting properties such as controlled and sustained release, low cytotoxicity, long-standing biomedical applications, tissue and cell biocompatibility, prolonged residence time and targeted delivery [133]. On this basis, it has been approved by the US Food and Drug Administration (FDA) and the European Medicines Quality Agency (EMA) as a superior drug carrier [134]. Nanoparticles have been investigated for the delivery of various antimicrobial agents, including AMPs (Figure 5), and efficacy in treating different types of infections has been reported [135]. D’Angelo et al. designed and developed a poly(lactic-co-glycolic acid) (PLGA) nanoparticle containing colistin (Col) by an emulsion/solvent diffusion technique. The Col-loaded NPs were found to kill *P. aeruginosa* biofilms and to display prolonged efficacy in biofilm eradication compared to free Col in a lung infection model [136]. The same nanocarrier system is used to encapsulate Esculentin-1a and its derivatives [137]. Prolonged efficacy against *P. aeruginosa* infections was demonstrated in vitro and in vivo, highlighting this system as promising for the local treatment of infectious diseases of the lung [138]. The SET-M33 peptide has also been encapsulated in poly(lactide-co-glycolic)(PLGA) nanoparticles [139]. Encapsulation of the peptide in PLGA-NPs conjugated with polyethylene glycol (PEG) strongly reduced the toxicity of the peptide in vitro and in vivo, maintaining its antibacterial effect [139].

## 8. Conclusions

The road to a new class of antibiotics is still long and rocky. When antimicrobial peptides (AMPs) attracted interest as a new class of possible antibiotics, research on them increased considerably. New peptides able to kill bacteria were discovered in plants, insects and higher animals, including humans [140], and were named Host Defence Peptides (HDPs). Later, bioinformatic procedures and combinatorial libraries were used to identify artificial sequences with antimicrobial properties and to modify the structure of known peptides. In the last 30 years, the structure, mechanisms of action, toxicity, pharmacokinetics and bio-distribution properties of AMPs and HPDs have been studied. A common feature that emerged was that all AMPs and HDPs initially interact with membranes but show relative toxicity, which narrows the therapeutic window (ratio of toxicity to efficacy), excluding clinical use of these molecules in an unmodified form. A few AMPs were approved for human treatment, and most are now considered drugs of last resort in cases of multi-drug-resistant bacteria. A former hypothesis that AMPs and HPDs did not select bacterial resistance because their mechanism of action involves membranes was refuted after a few years of clinical use, as in the case of colistin, to which an increasing number of bacteria are now resistant [90,91].

Despite all these considerations, the interest in AMPs as new antimicrobial drugs remains high because many are indeed effective, at least in vitro, against major pathogens with multi-resistant profiles for traditional antibiotics. The new impulse for research on these molecules concerns improved modification and optimisation strategies, innovative formulation approaches that enhance chemical and biological stability, encapsulation in new biocompatible materials for safe delivery and, finally, advanced chemical synthesis protocols aimed at lowering manufacturing costs. We are now entering a new phase of research aimed at formulating AMP molecular compositions for safe and effective use. In other words, after a long period of discovery and study of AMPs, the focus is now on their “druggability” through modifications and improvement of delivery systems.

In parallel with the development of AMPs as drugs, there has been recent interest in alternative applications, for example, in medical devices [141,142] and cosmetics [143]. We can expect interesting developments in the near future. 

## Figures and Tables

**Figure 1 ijms-25-04870-f001:**
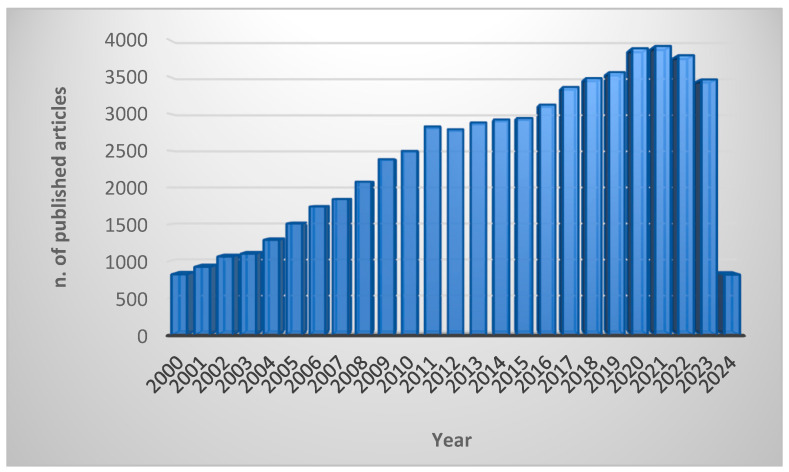
Number of scientific articles published in academic journals since 2000. The keywords used to search the PubMed database were “antimicrobial peptides” OR “AMPs” OR “host defence peptides” OR “HDPs”. The search was performed in March 2024.

**Figure 2 ijms-25-04870-f002:**
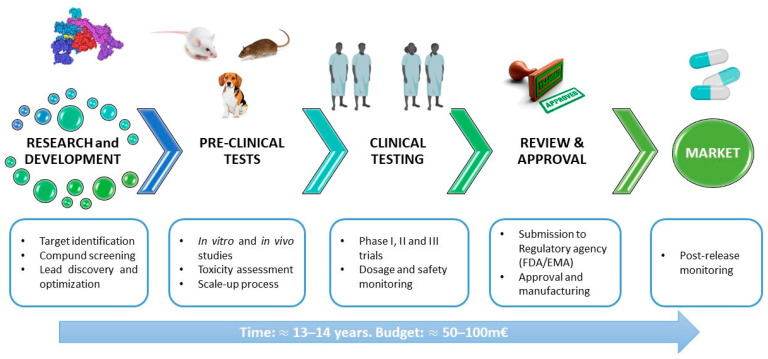
General scheme of the pipeline of antimicrobial drug development.

**Figure 3 ijms-25-04870-f003:**
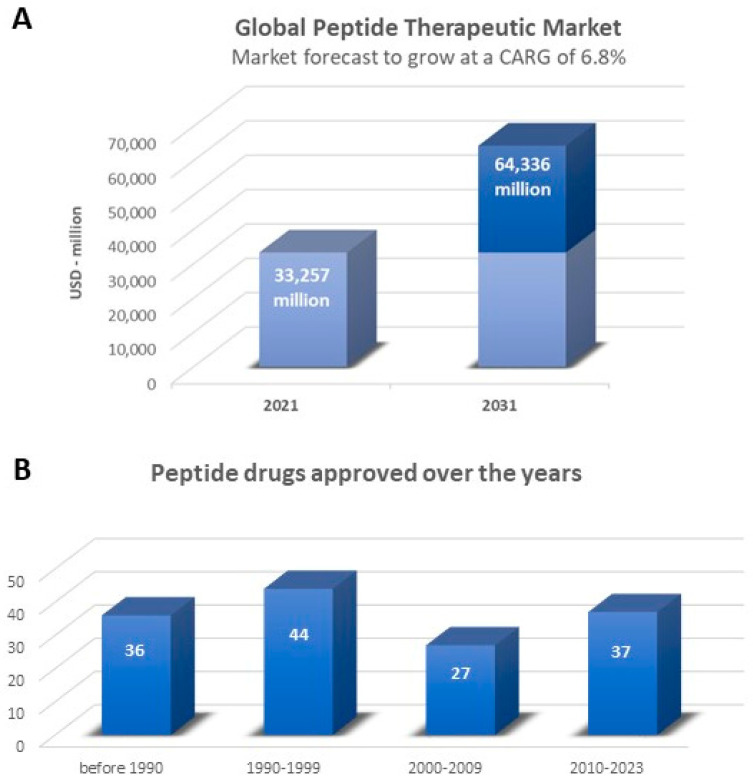
(**A**) Global peptide therapeutics market, 2021–2031 [53]. (**B**) Peptide drugs that have gained approval over the years [54].

**Figure 4 ijms-25-04870-f004:**
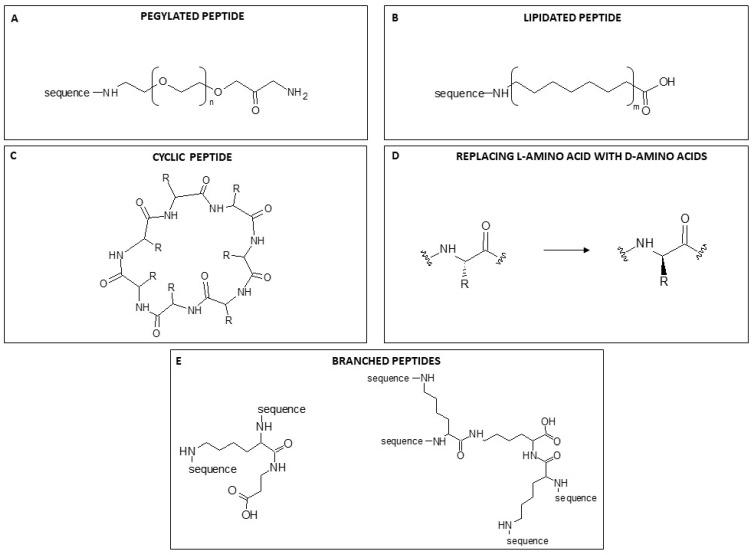
A general example of the chemical modification of a peptide structure. (**A**) PEGylation. *n* = 4–20 (**B**) Lipidation. *m* = -CH_2_- from 12 to 30. (**C**) Cyclisation. R = residue. (**D**). Replacement of L-amino acid with D-amino acids. R = residue. (**E**). General structure of a two-branched peptide synthesised on a one-lysine scaffold (on the left), and a tetra-branched peptide synthesised on a three-lysine scaffold (on the right). The proportions of structural components are not to scale.

**Figure 5 ijms-25-04870-f005:**
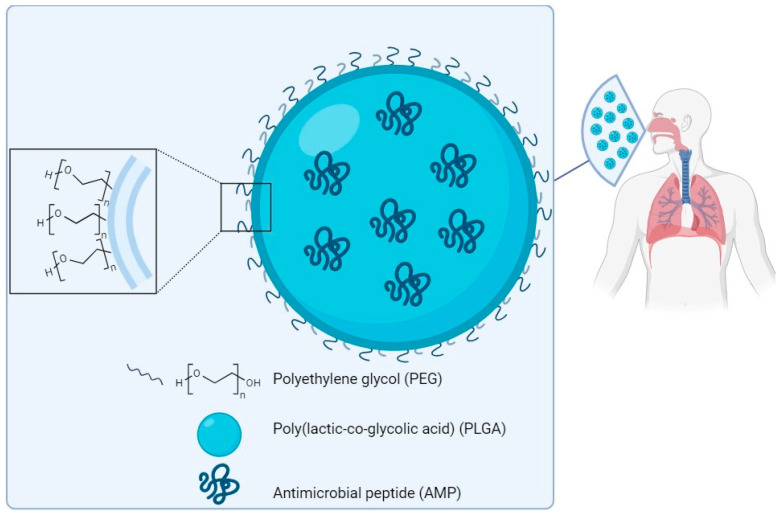
Scheme of a prototype of PLGA-PEG nanoparticles loaded with an antimicrobial peptide for delivery to the lung. Components not to scale.

## Data Availability

Data are contained within the article.

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
