# Peer review of "Antimicrobial Peptides towards Clinical Application—A Long History to Be Concluded"

_ijms, 2024, doi:10.3390/ijms25094870_

Round 1

Reviewer 1 Report

Comments and Suggestions for Authors

The review by Laura Cresti, Giovanni Cappello and Alessandro Pini reports on the development phase and clinical trials of antimicrobial peptides. The review is rather interesting but the text contains some minor flaws.

 Minor comments

 1.      Lines 252, 255, 266, 272 - name of organisms should be written in italics

 2.      Line 227  - it’s worth swapping the link [52] and (Figure 3C)

 3.      There is no mention of Figure 5 in the text (section 7). It might be better to replace the “diagram” in the caption to Figure 5 with a “scheme”

 4.       In my opinion, part describing de novo design of AMP (lines 113-121) should be expanded with examples of specific modeled peptides or specific achievements in this field obtained using AI.

Reviewer 2 Report

Comments and Suggestions for Authors

Dear Authors,

Thank you for the opportunity to review an interesting Review. Although the Manuscript is thoughtful and, for such a broad topic, written concisely it requires necessary corrections before publication:
Abstract:
"other biological events" is not the correct term. The action of AMP is known. Please write specifically what the authors meant, or change this sentence so that this term is not there.
"Limits" are also the huge costs associated with AMP production, which the authors should mention. This is one of the reasons why pharmaceutical companies are pulling out of participating in AMP production.
Introduction and the rest of the Article:
The introduction needs reworking.

Line 35 - are you sure the families are listed in the report?
I would ask the Authors to verify the information. In addition, the Authors wrote this part of the Introduction in such a way that the reader thinks that all the strains listed by the Authors are dangerous. Which is not true, because these are strains that exhibit certain, specific characteristics. Strain to strain is not similar, and the Authors' statement suggests that all Acinetobacter spp. are dangerous. Moreover, another point. We can't just use generic names in this case if specific species are used in the report. So, lines 33-38 need to be changed and should be expanded to include specific information about the pathogens that the report covers.

lines 51-57 not only those listed by the Authors.
There are others, such as meropenem with vaborbactam. Please rewrite this passage to include as examples the combinations given by the Authors, taking into account MER-VAB ( DOI: 10.3390/antibiotics12111612 ). The current form of the Authors' statement suggests that there are only those given by the Authors here.
Figures 2, 4 are unreadable. Too small or fuzzy, poorly visible captions.
"in vivo", "in vitro". - please change throughout the Manuscript to be written without italics.
"spp" or "ssp" should be written without italics. Please change throughout the Manuscript.
Line 295, 296 - when the name of the bacteria appears for the first time then the full name should be used, thus " Staphylococcus", "Klebsiella". Only on subsequent use can the abbreviation be used. This also applies to other species names from those used in this Manuscript (e.g. line 334 - C. difficile). Please review the entire Manuscript very carefully in this regard.

In my opinion, the above corrections will make the Manuscript more attractive.

Best regards

Reviewer 3 Report

Comments and Suggestions for Authors

Antimicrobial peptides towards clinical application – a long history to be concluded

Cresti et al.

It was a pleasure to read this manuscript. The review is well-organized and written. The figures are meaningful and make a significant contribution to the overall appeal of the manuscript to the readers both from scientific and general audiences. 

There are a few suggestions that I would like to give to the authors for consideration.

1. L. 60. "phage therapy, which uses viruses to target and destroy bacteria" This sentence is a little bit wordy. Please, consider rewriting.

2. L. 137-138. What about extracting and purifying antimicrobial peptides from bacteria? I understand that this is not that challenging in comparison to the extraction of antimicrobial peptides from animal and plant cells.

Extraction of endolysins from phage-infected bacterial cells also seems to be challenging. Please, consider rewriting this sentence.

3. I suggest adding one more aspect to the description of the data that can be obtained from pre-clinical testing using in vitro models. Please, check the Rolhion et al. study (https://doi.org/10.1016%2Fj.chom.2019.10.016). They used the artificial GI system (SHIME) to assess the effects of bacteriocin (Lmo2776) on the human gut microbiota composition in vitro. The addition of the description of the capabilities of using artificial GI systems, such as SHIME, TIM, ARCOL, etc. for the assessment of AMP's effects on complex gut microbial communities will improve the manuscript, as applying artificial GI systems with no doubt allows for the optimization of the drugs based on AMP (avoiding or minimizing usage of in vivo models).

4. L. 237 or 277. Please, consider adding some discussion of nisin, as this is the most studied bacteriocin, which is also considered AMP. Nisin is approved by regulators in over 80 countries, including the Food and Drug Administration (FDA) and the European Food Safety Authority (EFSA) (https://doi.org/10.1016/j.copbio.2017.07.011). Indeed this AMP is generally used as a biopreservative (https://doi.org/10.3390/foods11193145), but there are some clinical trials on applying nisin for the treatment of oral diseases (https://classic.clinicaltrials.gov/ct2/show/NCT06097468).

Please, consider adding a discussion about nisin to one of the paragraphs of your manuscript, as the absence of the discussion of this AMP in your review could confuse some readers, especially specialists in the field of AMP.

Reviewer 4 Report

Comments and Suggestions for Authors

In this review Pini and collegues describe the development phase and clinical trials of antimicrobial peptides (AMPs). They also provide an update on AMPs in the pharmaceutical industry and an overall view of their therapeutic market. Structure modifications to improve stability in vivo and bioavailability are also described.

The article is well organized and very easy to read.

Before publication I believe the authors should make some small changes/corrections listed below.

Figure 3: the figure as shown is unclear. In sections A and B the authors report the same data simply changing the time period taken into consideration. Authors are invited to choose between graphs A and B

Chapters 5 and 6: the authors are invited to add in these sections of the article some information regarding the mechanism of action of these AMPs

Check that the names of the (micro)organisms are in italics (e.g. line 238, 255, 266)

Round 2

Reviewer 2 Report

Comments and Suggestions for Authors

Dear Authors,

All changes made by the authors are acceptable. In my opinion, the Manuscript is suitable for publication in its present form. 

Best regards